# Targeting HIF-1α Function in Cancer through the Chaperone Action of NQO1: Implications of Genetic Diversity of NQO1

**DOI:** 10.3390/jpm12050747

**Published:** 2022-05-05

**Authors:** Eduardo Salido, David J. Timson, Isabel Betancor-Fernández, Rogelio Palomino-Morales, Ernesto Anoz-Carbonell, Juan Luis Pacheco-García, Milagros Medina, Angel L. Pey

**Affiliations:** 1Centre for Biomedical Research on Rare Diseases (CIBERER), Hospital Universitario de Canarias, 38320 Tenerife, Spain; esalido@ull.es (E.S.); ibetfer3@gmail.com (I.B.-F.); 2School of Pharmacy and Biomolecular Sciences, The University of Brighton, Brighton BN2 4GJ, UK; d.timson@brighton.ac.uk; 3Department of Biochemistry and Molecular Biology I, Faculty of Sciences and Biomedical Research Center (CIBM), University of Granada, 18071 Granada, Spain; rpm@ugr.es; 4Department of Biochemistry and Molecular and Cellular Biology, Faculty of Sciences, Institute of Biocomputation and Physics of Complex Systems (GBsC-CSIC Joint Unit), University of Zaragoza, 50009 Zaragoza, Spain; eanoz@unizar.es (E.A.-C.); mmedina@unizar.es (M.M.); 5Department of Physical Chemistry, University of Granada, 18071 Granada, Spain; juanlupacheco@correo.ugr.es; 6Unit of Excellence in Chemistry, Department of Physical Chemistry, Institute of Biotechnology, University of Granada, 18071 Granada, Spain

**Keywords:** HIF-1α, NQO1, hypoxia, angiogenesis, cancer, protein: protein interactions, ligand binding, proteasomal degradation, genetic variability

## Abstract

HIF-1α is a master regulator of oxygen homeostasis involved in different stages of cancer development. Thus, HIF-1α inhibition represents an interesting target for anti-cancer therapy. It was recently shown that the HIF-1α interaction with NQO1 inhibits proteasomal degradation of the former, thus suggesting that targeting the stability and/or function of NQO1 could lead to the destabilization of HIF-1α as a therapeutic approach. Since the molecular interactions of NQO1 with HIF-1α are beginning to be unraveled, in this review we discuss: (1) Structure–function relationships of HIF-1α; (2) our current knowledge on the intracellular functions and stability of NQO1; (3) the pharmacological modulation of NQO1 by small ligands regarding function and stability; (4) the potential effects of genetic variability of NQO1 in HIF-1α levels and function; (5) the molecular determinants of NQO1 as a chaperone of many different proteins including cancer-associated factors such as HIF-1α, p53 and p73α. This knowledge is then further discussed in the context of potentially targeting the intracellular stability of HIF-1α by acting on its chaperone, NQO1. This could result in novel anti-cancer therapies, always considering that the substantial genetic variability in NQO1 would likely result in different phenotypic responses among individuals.

## 1. HIF-1α: Structure, Function, Regulation and Disease

Oxygen homeostasis is one of the main principles of life for most organisms. The availability of oxygen regulates many physiological processes that are strictly controlled by a complex network of molecules in the organism, with hypoxia-inducible factors (HIFs) playing a central role. These transcription factors are heterodimers composed of an alpha (HIF-α) and a beta subunit (HIF-β). There are three isoforms of HIF-α: HIF-1α, HIF-2α and HIF-3α. Both HIF-1α and HIF-2α regulate the expression of a multitude of genes in response to oxygen availability and play a fundamental role in many physiological and pathological processes [1,2,3]. These two isoforms share structural features and some functional roles but they also have unique, sometimes opposing activities [3]. Since the focus of this review is the chaperone effect of NQO1, which has been studied on HIF-1α, we will mostly refer to this isoform.

### 1.1. HIF-1α Structure and Function

HIF-1 is a DNA-binding complex composed of two subunits (α and β), both basic helix-loop-helix (bHLH) proteins of the PER-ARNT-SIM (PAS) family. HIF-1β (initially named aryl hydrocarbon nuclear receptor translocator, ARNT) is constitutively expressed and, thus, its expression and activity are oxygen-independent. In contrast, HIF-1α is strongly repressed in normoxia and induced in hypoxia [4]. In response to low oxygen, HIF-1α dimerizes with HIF-1β, and HIF-1 binds to the hypoxia response element (HRE) of the promoter or enhancer region of the target genes to induce transcription. HIF-1 induces the transcription of numerous genes [5,6] involved in many biological processes, such as angiogenesis, erythropoiesis, anaerobic metabolism, cell survival, and cell proliferation.

The basic domain and the carboxy-terminus of PAS are specifically required for the DNA binding of HIF-1 to the HRE DNA sequence 5-‘RCGTG-3’, whereas the HLH domain and the amino-terminus of the two subunits of PAS proteins mediate their dimerization [7]. In addition to ubiquitous HIF-1α and HIF-1β, both subunits exist as a series of isoforms encoded by distinct genetic loci with different splice variants [8,9,10,11]. HIF-1α constitutes the most prominent member of the three HIF-α subunits in the human genome. The most important structural elements of HIF-1α are summarized in Figure 1A. HIF-1 has two nuclear location sequences (NLS) at the amino (residues 17–74) and carboxyl (residues 718–721) ends of the protein [12]. The stability of the protein is mainly regulated at the oxygen dependent degradation domain (ODDD). This domain contains two peptide sequences rich in proline (P), glutamic acid (E), serine (S), and threonine (T) at residues 499–518 and 581–600 [4]. PEST-like motifs are present in short-lived proteins rapidly targeted for intracellular degradation. They mediate HIF-1 degradation by the ubiquitin-proteasome pathway [13,14]. HIF-1α also contains two transactivation domains, N-TAD (residues 531–575) and C-TAD (residues 786–826) [15]. Under hypoxia, several co-activators such as CBP/p300/SRC-1 interact with the C-TAD inducing HIF-1α transcriptional activity, while N-TAD is responsible for stabilizing HIF-1α against degradation [16]. 

### 1.2. Regulation of HIF-1α Expression

Although HIF-1α is tightly regulated at the transcriptional, translational and posttranslational levels [5], protein degradation plays an important role in regulating its activity and is the main topic of this review, focused on the interaction between NQO1 and HIF-1α. 

#### 1.2.1. Oxygen-Dependent Regulation

Most of HIF-1 regulation is mediated by the HIF-1α subunit. The best known mode of HIF-1α regulation, and more related to the topic reviewed here, takes place at the postranslational level, where oxygen regulates HIF-1α stability. Under normoxic conditions, HIF-1α protein is rapidly targeted for ubiquitination and proteasomal degradation, thus keeping minimal steady-state levels. In this scenario, HIF-1α becomes hydroxylated using O_2_ as substrate by members of the prolyl hydroxylase domain (PHD) family on two specific prolines (P402 and P564) of the ODDD and NTAD domains, respectively [17,18,19]. Hydroxylation of these proline residues generates a binding site for an E3-ubiquitin ligase complex that contains the von Hippel Lindau protein (VHL), elongin B (ELB), elongin C (ELC), cullin 2 (CUL2), and ring box protein 1 (RBX1). As a result, HIF-1α is polyubiquitinated and degraded by the proteasome [20,21,22,23] (Figure 1B). Oxygen also affects DNA binding and transcriptional activity. This effect is mediated by the factor inhibiting HIF-1α (FIH-1), an asparagyl hydroxylase that catalyzes the hydroxylation of Q803 within the C-TAD domain of HIF-1α. Hydroxylation of Q803 leads to a steric clash that prevents the binding of the p300 and CBP coactivators to HIF-1α, abolishing their transactivation [24].

In addition, *HIF1A* transcription is also regulated [25] and more recent evidence indicates that the regulation of *HIF1A* mRNA translation is a crucial element of HIF1α induction under hypoxia and stress conditions, making the translational machinery an important target for cancer therapy (reviewed in [26]). 

In hypoxia, impaired hydroxylation of HIF-1α allows dimerization of the transcriptional complex and the subsequent expression of its target genes. The mitochondrial electron transport chain (MEC) consumes the vast majority of cellular oxygen and may play an important role in HIF-1α regulation by determining the amount of oxygen available to PHD enzymes [27,28,29]. In this sense, pharmacological or genetic inhibition of mitochondrial respiration may prevent the accumulation of HIF-1α in hypoxia [30,31]. However, there are other ways in which mitochondria affect the stability of HIF-1α (Figure 1B). Thus, in addition to requiring 2-oxoglutarate as a co-substrate, several metabolites of the tricarboxylic acid (TCA) cycle, especially succinate and fumarate, additionally inhibit the HIF hydroxylases [29,32,33,34]. Reactive oxygen species produced by MEC in hypoxia also increase the stability of HIF-1α [30,35,36,37,38]. 

#### 1.2.2. Growth Factor Signaling Pathways

Growth factors, cytokines and hormones [39,40,41,42,43,44] also regulate HIF-1α. These signals lead to the activation of mTOR (by the PI3K/AKT pathway) and the mitogen activated protein kinases (MAPK) (by the RAS/RAF/MEK/ERK kinases cascade) which can phosphorylate 4E-BP1, favoring translation of *HIF1A* mRNA [45,46,47]. The translational regulation of *HIF1A* mRNA has been recently reviewed [25]. MAPKs also play an important role in the regulation of the transactivation activity of HIF-1. Once activated, members of this family phosphorylate HIF1α, enhancing the binding of the coactivator p300/CBP [46,48].

#### 1.2.3. The Mdm2 Pathway

A complex interplay exists between p53 and HIF-1α. On one hand, HIF-1α binds to Mdm2 (the mouse double minute 2 homologue), inhibiting Mdm2-dependent degradation of p53 [49]. On the other hand, p53 also mediates HIF-1α activity. In this sense, at moderate expression levels, p53 competes with HIF-1α for p300, consequently decreasing the transcriptional activity of HIF-1α. Moreover, high levels of p53 expression lead to HIF-1α degradation [50]. This is due to the binding of p53 to HIF-1α that allows Mdm2-mediated ubiquitination of HIF-1α and its proteasomal degradation [51] (Figure 1B). 

#### 1.2.4. Heat Shock Protein 90 (Hsp90)

HIFα stability is also regulated by other signaling pathways, since Hsp90 inhibitors promote HIFα degradation in a pVHL-independent manner [52,53]. In addition, Hsp90 is known to bind directly with HIF-1α, inducing some conformational changes in its structure that improve the fitting and coupling with HIF-1β, thus initiating its transactivation [54] (Figure 1B).

### 1.3. HIF-1α in Cancer

Cancer is defined as the autonomous growth and spread of a clone of somatic cells, under selective pressure, which is able to use multiple cellular pathways to modify its environment in favor of its own proliferation and against the needs of the organism, escaping physiological constraints on cell growth, including immune surveillance [55]. 

As the cell clone grows, its core becomes further removed from the blood supply and changes in oxygen availability are key for tumor cells to reorganize their metabolism in order to match oxygen supply with cellular demands. Thus, hypoxia becomes one of the key features of the tumor microenvironment [56], a fact first reported in early studies where hypoxic cells were shown to surround the necrotic center in histological sections of lung carcinomas [57] and later was demonstrated in essentially all solid tumors (reviewed in [56]). Intratumoral hypoxia is widespread and almost independent of tumor size, grade, stage, or histology [58]. Numerous studies have elucidated the molecular mechanisms involved in the cellular response to hypoxia, and three leaders in the field have been awarded the 2019 Nobel Prize in Medicine for their achievements, including their seminal work on HIF-1α, a critical transcription factor for adaptation to hypoxia [59,60] (https://www.nobelprize.org/prizes/medicine/2019/summary/, accessed on 1 April 2022).

HIF-1α and HIF-2α regulate a number of physiological pathways involved in cancer, such as cell proliferation, survival, apoptosis, angiogenesis, glucose metabolism, immune cell activation and stem cells [59,60,61]. In particular, they play an important role in at least two of the main biological capabilities acquired during the multistep process of tumor development (the “hallmarks” of cancer [55]): reprogramming cell metabolism and inducing angiogenesis.

The complexities of cancer prognosis (the high variability of tumor mutations, histological types, grades and stages) make it too simplistic to think that the levels of expression of a single gene such as *HIF1A* might have a significant impact on prognosis. However, increased HIF-1α and HIF-2α protein levels in diagnostic tumor biopsies have been correlated with poor prognosis in a variety of cancer types (reviewed in [1]). Recently, high HIF-1α expression has been studied in triple-negative breast cancer (TNBC) [62]. TNBC are a heterogeneous group of poorly differentiated, highly aggressive and metastatic breast cancer type. They lack specific molecular-targeted therapy and there is an urgent need to find effective biomarkers rather than relying of their lack of hormone receptors and Her2 expression to define this subgroup. These authors [62] found that HIF-1α and c-Myc expression detected by immunohistochemistry could add to the development of a prognostic nomogram together with the histological grade and stage of cancer (based on tumor size (T), lymph node involvement (N) and distant metastases (M), known as TNM status). In an effort to validate the proposed nomogram in our institution, we are currently evaluating the feasibility of including HIF-1α immunohistochemistry in the biomarker profile (Figure 2). As transcription factors induced in tumor cells by hypoxic environmental stress, HIF-1α and HIF-2α could regulate a variety of downstream genes, thereby increasing the invasive ability of tumors and their resistance against standard radiotherapy and chemotherapy. 

## 2. NQO1 as a Protein Chaperone: Current Knowledge and Potential Application to Target HIF-1α Stability 

The description of NQO1 as a protein chaperone protecting HIF-1α against degradation [63] supports that targeting this protein–protein interaction (PPI) could allow inactivation of HIF-1α. In this section we will describe a wealth of information available for the activity and stability of NQO1, focusing on what we know about other PPI, by which NQO1 acts as a chaperone and how small ligands and missense mutations are capable of modulating these PPI. We hope that this information will boost interest in targeting NQO1 as a modulator of HIF-1α.

### 2.1. Overview of NQO1 Expression, Regulation and Functions: On the Potential Roles of O_2_ Levels and HIF-1α

NAD(P)H quinone oxidoreductase 1 (NQO1; DT-diaphorase; EC 1.6.5.2) functions as a homodimer of 62 kDa [64,65]. Each subunit contains two domains: an N-terminal domain (NTD, comprising approximately residues 1–225), which tightly binds one molecule of FAD and is capable of folding and assembly into dimers autonomously, and a C-terminal domain (CTD, comprising the last 40–50 residues) that completes the monomer:monomer interface (MMI) and the binding sites of NAD(P)H and the substrates (Figure 3A) [64,66,67,68,69,70]. Its main enzymatic function is to catalyze the two-electron reduction of quinones, thus avoiding the production of reactive, and potentially cytotoxic, semiquinones (Table A1) [71]. Since one of its substrates is menadione (vitamin K3) it may function in blood clotting, although its role appears to be minor compared to vitamin K oxidoreductase [72]. Other substrates include α-tocopherol-quinone and ubiquinone, which are maintained in their reduced (antioxidant) forms by the enzyme [73,74]. NQO1 can also catalyze the reduction of superoxide radicals and thus plays a role in directly protecting the cell from reactive oxygen species (ROS) [75]. 

The enzymatic cycle of NQO1 follows a ping-pong mechanism with two main steps or semi-reactions: in the first one, the rate-limiting step, NAD(P)H binds to the holo-form (NQO1_holo_) of the enzyme (FAD is not released during the cycle) [65]. Consequently, NAD(P)H reduces the flavin to FADH_2_ and the NAD(P)^+^ is released. In the second half-reaction, a two-electron reduction occurs between FADH_2_ and the substrate (Q). However, the efficiency of catalytic sites in the dimer, either in the reductive or oxidative half-reactions, has been shown to be non-equivalent [76,77]. Steady-state and pre-steady-state studies have supported the presence of a substantial functional negative cooperativity in the NQO1 catalytic cycle (even in the FAD binding affinity) [76,78,79]. Essentially, one of the active sites operates one-to-two orders of magnitude faster than the other one, indicating the existence of strong allosteric communication between the active sites during the catalytic cycle [76,77] (Figure 3B). We must note that the rate of this reductive half-reaction is strictly NAD(P)H concentration dependent, and thus, should be accelerated due to the HIF-mediated enhancement of aerobic glycolysis (raising cytosolic NADH levels). In the second and faster oxidative half-reaction [76], the substrate binds and is reduced by the FADH_2_, thus releasing the reduced substrate and regenerating the holo-enzyme [65,68]. Dicoumarol, a potent anticoagulant and mitochondrial uncoupling agent, acts as a competitive inhibitor of NAD(P)H and the substrate during the full NQO1 cycle (Table A1) [76,80]. The high level of NQO1 expression in some cancer cells, coupled with its role in defense against ROS, has led to the proposal that NQO1 inhibition may be a novel and plausible therapy for this disease. Actually, dicoumarol is an effective inhibitor of pancreatic cancer cells in culture, but no NQO1 inhibitors are yet in clinical use [81]. However, the situation is far from simple, and it has also been shown that NQO1 promotes AMPK activation and induces cancer cell death under oxygen–glucose deprivation [82].

**Figure 3 jpm-12-00747-f003:**
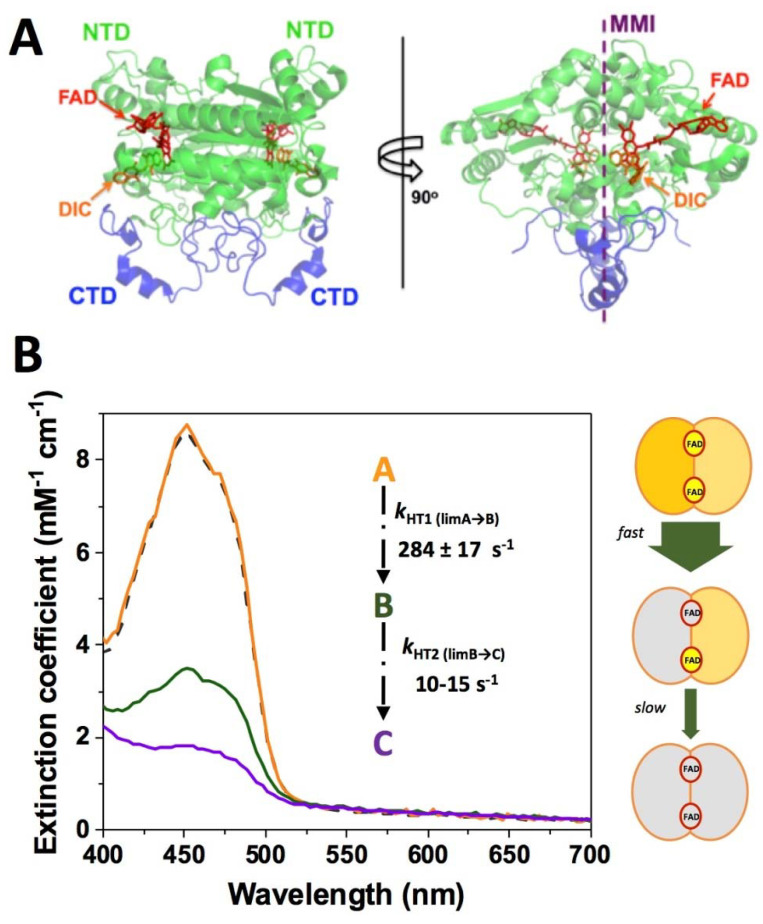
Structure and catalytic function of NQO1. (**A**) Structural representation of the NQO1 dimer (PDB 2F1O) [83]. NTD and CTD refer to N-terminal and C-terminal domains, respectively. The location of the FAD and the inhibitor dicoumarol (DIC) binding sites is also indicated. The monomer:monomer interface is indicated as MMI. (**B**) Reduction of FAD by NADH shows two different pathways. In the left panel, the spectral properties of the different spectroscopic species (A, B, C) stabilized upon reduction are indicated as well as their conversion limiting rate constants. The right panel shows the model proposed by us for the sequential reduction of the two FAD cofactors in the protein homo-dimer, in which this large difference in kinetics represents a type of functional negative cooperativity. Adapted from [76].

The NQO1 protein also has non-enzymatic functions in which it mostly participates in PPI, modulating the stability of its binding partners (Figure 4 and Table A2). Indeed, and as we will further discuss in Section 2.2 and Section 2.3, several of the ligation states populated by NQO1 during the catalytic cycle (NQO1_holo_, either oxidized and reduced, or the *dead-end* complex of the holo-enzyme with dicoumarol, NQO1_dic_) are very relevant to the discussion of PPIs developed by NQO1. Among them, the reduced form of NQO1 seems to be most critical for strengthening PPI as well as the change in the subcellular location of the enzyme [84,85,86]. From a (bio)chemical point of view, its intrinsically low kinetic stability (i.e., a very small half-life, in the order of a few ms for the reduced flavin state) in the presence of model substrates is intriguing [76]. However, the results from Ross and coworkers have additionally supported that from a cellular/physiological viewpoint; this NQO1 state with the flavin reduced could be of high relevance, locally controlling the NADH/NAD^+^ ratio, the cellular location of NQO1 and α-tubulin organization [85,86].

NQO1 is known to undergo different post-translational modifications (PTMs), some of which have been characterized in a site-specific manner. These studies have revealed that phosphorylation may modulate NQO1 function and stability in different manners. Phosphorylation of NQO1 at T128 by the Akt kinase results in polyubiquitination and proteasomal degradation of NQO1 [87]. A phosphomimetic mutation on the residue S82 showed reduced FAD binding affinity and consequent loss of intracellular protein stability, although the kinase involved in this phosphorylation event has not been identified yet [88]. Therefore, it is reasonable to propose that phosphorylation of NQO1 described at multiple sites (https://www.phosphosite.org/proteinAction.action?id=14721&showAllSites=true, accessed on 1 April 2022) may have an impact on the intracellular stability of its protein partners through the destabilization of NQO1. 

*NQO1* is expressed in most human cell types (https://www.genecards.org/cgi-bin/carddisp.pl?gene=NQO1, accessed on 1 April 2022) and its expression can be upregulated in response to stress. Two control elements are located upstream of the human *NQO1* gene: the antioxidant response element (ARE) and the xenobiotic response element (XRE) [89,90,91,92]. A key regulator of the ARE is nuclear factor erythroid 2-related factor 2 (Nrf2). This transcription factor from the basic leucine zipper family is located in the cytosol under non-stressed conditions. Here it interacts with Kelch like-ECH-associated protein 1 (KEAP1) that targets it for ubiquitin-mediated proteasomal degradation [93,94]. Oxidative stress is sensed by the oxidation of disulfide bonds in KEAP1 [95]. This releases Nrf2, which translocates to the nucleus and, together with members of the small musculoaponeurotic fibrosarcoma protein family (sMaF), binds to AREs [96]. This results in the recruitment of the mediator complex, the acetylation of Nrf2 and the activation of a range of genes involved in antioxidant defense, including NQO1 [97,98,99]. Nrf2 also upregulates genes for NADPH generation, thus ensuring that enzymes such as NQO1 have sufficient reduced cofactor [100,101]. The ARE also responds to some antioxidants (e.g., resveratrol and quercetin), by upregulating the expression of *NQO1* and other antioxidant enzymes [102,103]. Interestingly, the ARE that controls the expression of *NQO1* is also activated by hypoxic conditions [104]. Thus, NQO1 protein levels increase under hypoxic conditions [105]. Consequently, *NQO1* is overexpressed in tumors due to their hypoxic environment.

The XRE is activated by the aryl hydrocarbon receptor (AhR), a basic helix-loop-helix transcription factor. This protein binds to a wide range of hydrophobic xenobiotics and translocates to the nucleus in response. Here, it heterodimerizes with aryl hydrocarbon nuclear translocator (ARNT or HIF-1β) and binds the XRE [106,107]. In the case of the *NQO1* gene, AhR at the XRE and Nrf2 functionally cooperate to regulate its expression [108]. Like the ARE, the XRE can be induced by hypoxia [109]. Although HIF-1α also partners with HIF-1β to regulate transcription, siRNA silencing of the *HIF1A* gene does not affect NQO1 expression [110].

**Figure 4 jpm-12-00747-f004:**
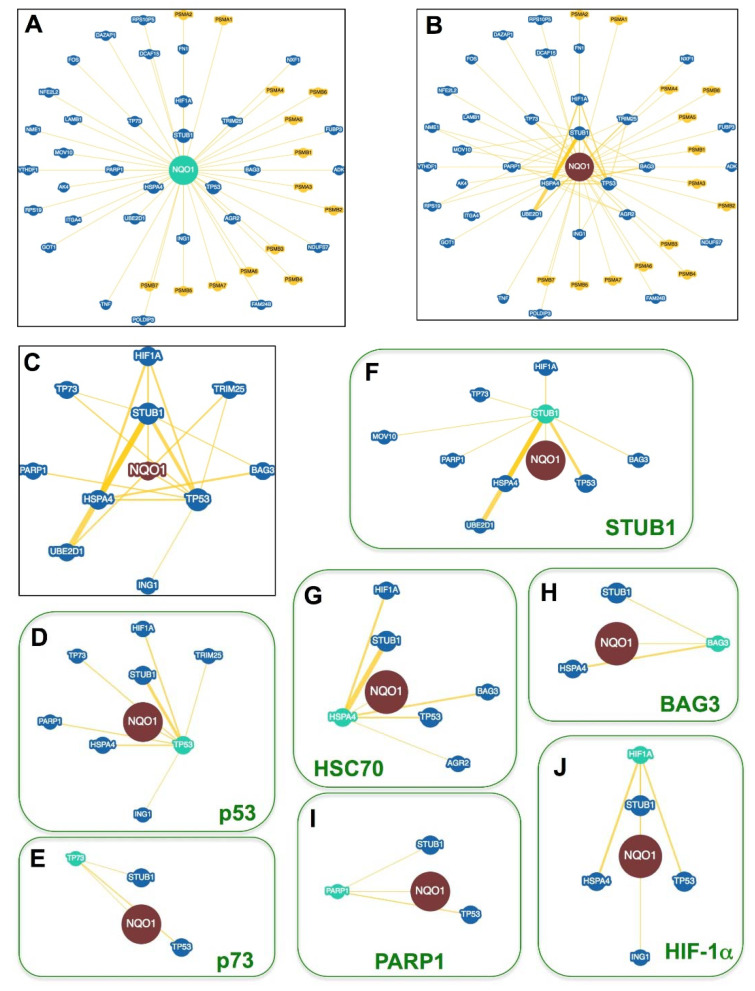
The NQO1 interactome. (**A**) NQO1 protein partners described in at least one report. (**B**) Connectivity between NQO1 partners. The shorter the radial distance between a partner and NQO1 indicates a higher interconnectivity. Partners in yellow are from mouse and in blue from human. (**C**) A zoom from panel (**B**) shows the most highly interconnected partners of NQO1. (**D**–**J**) Interactions between NQO1 partners in panel (**C**) and other proteins to highlight their different interconnectivity. The thickness of connecting yellow lines is related to the number of reports describing the interaction. Data were retrieved from the BioGRID database (https://thebiogrid.org/108072, accessed on 1 April 2022) [111].

### 2.2. NQO1 Macromolecular Interactions

To date, there are 47 protein partners of NQO1 compiled in the BioGRID database (https://thebiogrid.org/108072, accessed on 1 April 2022) [112]. This set includes thirty-three human proteins (actually one is a pseudo-gene product). The remaining fourteen are from rodents and belong to different subunits of the 20S proteasome (Figure 4A). In addition, NQO1 can interact specifically with certain RNAs modulating their expression [113]. Focusing on protein partners, there is certain evidence within the NQO1 interactome supporting that different partners (i.e., nodes) of this network can interact not only with NQO1 but also among each other. Consequently, this could generate regulatory circuits upon the interaction of different partners in two- or multiple-way manners (i.e., binary, ternary, quaternary, etc.…complexes; Figure 4B). This also suggests that we may affect the properties (e.g., intracellular activity or stability) of one node by targeting different adjacent nodes in the network, an effect that could be amplified particularly in nodes with high connectivity. The most representative and consistently reported nodes with such a high connectivity for NQO1 are shown in Figure 4C and Table A2. Importantly, HIF-1α belongs to this group. 

A critical point to achieve a better understanding of the functional consequences of NQO1 interactomics emerges from a well-known effect. This is the interaction of NQO1 with protein partners that enhances the stability of these partners against both Ub-dependent or –independent proteasomal degradation [63,114,115]. Thus, effects on the intracellular levels of NQO1 or in the strength of its interaction with partners should translate into altered stability of the nodes in the NQO1 interactomic network. For instance, reducing the availability of the FAD precursor riboflavin leads to an increased population of the flavin-free NQO1_apo_ state, which is highly sensitive to proteasomal degradation through ubiquitin-dependent and -independent mechanisms [116,117]. This is in part caused by the destabilization of the CTD of NQO1 in the absence of bound FAD that promotes its ubiquitination and degradation [117]. Remarkably, addition of the inhibitor dicoumarol does not stabilize WT NQO1 protein intracellularly and seems to reduce its ability to interact with and stabilize these protein partners (see Table A2). The binding of NAD(P)H, which reduces the bound flavin cofactor, seems to enhance the interaction of NQO1 with these protein partners ([84], Table A2). A recent study also suggested that NQO1 with a reduced flavin cofactor could be a physiologically important state in the cell [85], and thus very relevant to understand the interactome dynamics of NQO1 (note that the population of this state can be promoted due to HIF-mediated increased levels of cytosolic NADH). Additional factors modulating NQO1 intracellular stability, such as protein phosphorylation events, small ion binding or single amino acid changes [111,118,119,120,121], may thus propagate and amplify these effects to the stability of the NQO1 protein partners.

Another critical issue for understanding the interaction of NQO1 with its partners and subsequent effects of these interactions on the stability of its interactomic consequences is the subcellular location of NQO1 and its partners. It is generally accepted that NQO1 subcellular location is primarily cytosolic, where it can fulfill its roles as antioxidant/detoxifiying enzyme and is able to interact with protein and RNA partners. However, there is also certain evidence supporting its location in other subcellular compartments, mostly mitochondria and the nucleus (with low confidence in other organelles) (Figure 5A; GeneCards^®^; https://www.genecards.org/cgi-bin/carddisp.pl?gene=NQO1&keywords=NQO1, accessed on 1 April 2022). Importantly, most of the protein partners of NQO1 also seem to primarily localize in the nucleus and the cytosol, and to a moderate extent in the mitochondria, cell membrane and extracellularly, apparently coexisting between more than one subcellular location (Figure 5A,B). In the case of the cytosol and the nucleus, about 50% of NQO1 partners may operate in any of these locations and translocate between these two to fulfill their roles in cytosolic and nuclear processes such as the control of DNA expression (p53, HIF-1, c-FOS, NRF-2, NME1, FUBP3), the regulation of RNA and translation (POLDIP3, YTHDF1, NXF1, MOV10, DAZAP1, RPS19), the regulation of nuclear protein function (PARP1 and TRIM25), protein folding and degradation (BAG-3, eIF4G1, DCAF15, HSPA4, STUB1 and UBE2D1). In addition, a number of metabolic enzymes in the cytosol and mitochondria (ODC, NDUSF7, NME1, ADK, GOT1 and AK4) could also interact with NQO1 in these compartments. Indeed, NQO1 overexpression in mice is known to enhance glycolytic and mitochondrial respiration activities and enhance metabolic flexibility, mimicking the beneficial effects of caloric restriction [122]. It is worth noting that the proteasomal protein degradation machinery may operate through rather similar mechanisms in the cytosol and the nucleus [123,124], and thus, the well-known chaperone role of NQO1 for different protein partners (see Table A2) may indeed operate in the cytosol, thus increasing the levels of cytosolic proteins amenable to importing to other organelles (such as nucleus or mitochondria [63,125]) and plausibly by stabilizing these proteins upon import of both the partner and NQO1 in these organelles. It is noteworthy, although more rarely described, that the presence of NQO1 in other subcellular locations such as the cytoskeleton may explain other roles of the multifunctional NQO1 protein. For instance, recent studies have supported that NQO1 may provide an adequate supply of NAD^+^ for the deacetylase activity of different sirtuins associated with microtubule dynamics [85,122,126]. These studies also highlight the potential plasticity of NQO1 subcellular location during different cellular conditions or stages (e.g., the recruitment of cytosolic NQO1 to cytoskeletal structures during cell division) [85,86].

### 2.3. Changes in NQO1 Stability, Structure and Dynamics upon Mutation and Ligand Binding: Implications for the Stability of Its Protein Partners

The functional chemistry of NQO1 is remarkably complex [65,127]. At least four different ligation/redox states (with small molecules bound or oxidoreductive states of the flavin) are relevant to understanding the intracellular stability and enzymatic activity of NQO1. Similarly, these states have different functional consequences as well as effects on its macromolecular partners: NQO1_apo_, which has no ligand bound; NQO1_holo_, which contains a molecule of oxidized FAD per NQO1 monomer; NQO1_holo-red_, containing the FAD cofactor reduced upon interaction with NAD(P)H and ready for hydride transfer to the substrate; NQO1_dic_, a ternary complex of NQO1_holo_ with the inhibitor dicoumarol bound that competitively inhibits NAD(P)H and substrate binding. These different ligation states interact differently with NQO1 protein partners [67,84,125]. High-resolution structural models for these complexes are, to the best of our knowledge, not reported. Thus, we will focus in this section on the structural and dynamic consequences of ligand binding to NQO1 in an attempt to obtain some insight into their regulatory effects on NQO1 interactions with protein partners (always from an NQO1 point of view). NQO1_apo_ exists as a stable and expanded dimer characterized by significantly large conformational flexibility [65,67,68,69,111,128,129,130,131]. Although no high resolution structural model is available for this state, recent kinetic studies using hydrogen–deuterium exchange mass spectrometry (HDXMS) have identified a minimal stable core that holds the protein dimer, while most of the protein exists forming a highly dynamic structural ensemble, including the FAD and substrate binding sites in non-competent states for binding [132]. The local stability of this state is very sensitive to mutations [131]. This remarkable conformational flexibility makes NQO1_apo_ likely the most relevant state to understand the intracellular stability of NQO1, since the flexible CTD acts as an initiation site for rapid degradation through ubiquitin-dependent and -independent proteasomal pathways [88,117,131]. It is plausible that this unstable and highly dynamic NQO1_apo_ state is not capable of interacting with NQO1 protein partners, or at least, does so with lower affinity [67].

Binding of FAD triggers a large conformational change leading to compaction of the protein dimer, an increase in ordered secondary structure, overall conformational stabilization and a large decrease of protein dynamics that is sensed in almost the entire protein structure [67,68,69,78,111,128,129,131,132]. This NQO1_holo_ state is amenable for high-resolution structural studies [70] (Figure 6A). This state is also known to be intracellularly stable and likely to interact with protein partners quite efficiently [85,111,117,118,122]. 

The binding of NAD(P)H results in the FAD reduction, leading to a state we may name NQO1_holo-red_. This reaction is very fast (Figure 3) [68,76] and this state is likely unstable unless strongly reducing and/or anaerobic conditions are used (at least in vitro [68,76,86]). Consequently, detailed structural analyses of this state are not yet available. However, it is worth commenting on some studies carried out under less stringent (i.e., aerobic) conditions, particularly because a wealth of data support that, generally, this state may strengthen the interaction of NQO1 protein with protein partners [84,125]. Using biochemical in vitro assays with NAD(P)H, it has been reported that NQO1_holo-red_ influences the conformation of the CTD affecting the interaction with antibodies raised against it [85], apparently increasing the thermodynamic stability of this domain [69,85] and targeting the protein to the microtubules [86]. Intriguingly, the changes in conformation and stability of the CTD reported for NQO1_dic_ and NQO1_holo-red_ are strikingly similar, suggesting again that some subtle differences in structure and dynamics between these two states are likely responsible for their opposing effects on the interaction of NQO1 with protein partners.

The binding of dicoumarol (or NAD^+^) causes small structural rearrangements in the protein structure [64,67,70,83,129] (Figure 6A). Therefore, from an NQO1 structural point of view, it is not straightforward to understand how NQO1_dic_ prevents binding to or largely decreases binding affinity for macromolecular partners with subsequent effects on the stability of these partners [67,125]. An alternative explanation has emerged from studies on NQO1 protein dynamics and local stability by HDXMS [132]. The binding of dicoumarol to wild-type (WT) NQO1 causes strong effects on NQO1 protein dynamics, affecting the local stability of the protein core, and these effects propagate through long distances in the protein structure (Figure 6B). Some of these effects are sensed by the CTD (Figure 6B), which may explain how dicoumarol binding mildly increases the thermodynamic stability of this domain by about 1.5 kJ·mol^−1^ [69,131]. Interestingly, a significant fraction of the residues whose stabilities are affected by dicoumarol binding appear on the protein surface and far from the ligand-binding site (Figure 6C). This may suggest that NQO1_dic_ interacts differently with macromolecular partners through changes in the local stability and dynamics of this (these) binding site(s), which are likely located on NQO1 protein surface.

### 2.4. Mutations and Polymorphisms in NQO1: Disease and Protein Interactions

The association of NQO1 activity with several human diseases having a huge social impact, particularly cancer, HIV infection and neurological and cardiovascular diseases, has attracted attention on the effects of naturally-occurring single amino acid changes on NQO1 multifunctionality and the potential predisposition provided by these changes towards disease development [65,68,78,111,117,120,121,128,133,134,135,136,137,138]. There were 106 missense variants described in the human population (ExAC and gnomAD databases; https://gnomad.broadinstitute.org/gene/ENSG00000181019?dataset=gnomad_r2_1, 1 January 2021) and 47 missense variants in the COSMIC database (https://cancer.sanger.ac.uk/cosmic/gene/analysis?ln=NQO1#variants, 1 January 2021). By far, the two most common single amino acid variants are p.P187S and p.R139W, with allelic frequencies in the human population of about 0.25 and 0.03, respectively. Consequently, these two polymorphisms have been characterized in a large detail [67,68,78,111,118,121,128,129,139]. Additionally, over 25 mutations in *NQO1*, including rare natural variants (i.e., from gnomAD or COSMIC databases), evolutionarily divergent mutations and artificial mutations have been characterized in vitro and in cellulo. These mutations were originally (but not exclusively) aimed at characterizing the structural propagation of stability effects across the NQO1 structure. Therefore, the mutations may potentially affect different functional sites to different extents [65,120,129,131,133,134,140]. Overall, these studies strongly supported that mutational effects (either from natural, designed or evolutionary-derived mutations) can be transmitted to long distances (over 40 Å within the NQO1 dimeric structure, a distance that represents basically the entire size of the NQO1 monomer), and that these effects are strongly dependent on the ligation state. Actually, we have recently demonstrated these are long-range at high resolutions with HDXMS on p.S82D and p.P187S mutants [131]. It is known that the protein:protein interactions (PPIs) developed by NQO1 may be weakened when the inhibitor dicoumarol is bound to NQO1 [67,84] and, fortunately, high resolution structural and energetic information for this NQO1 state is available [83,132]. However, this type of information is not available for the NQO1 state with the flavin reduced, which is likely essential to understanding some physio-pathological roles of NQO1 [85,86,127].

The intracellular effects of p.P187S, the most common polymorphism in NQO1, can be ascribed to changes in both stability and activity. This polymorphism decreases, by 10- to 40-fold, the affinity for FAD thus favoring the population of NQO1_apo_ [68,78,79,118,120,133]. Unlike WT NQO1, p.P187S, in both NQO1_holo_ and NQO1_apo_, is degraded similarly due to a large thermodynamic destabilization of the CTD that triggers ubiquitination and subsequent proteasomal degradation [67,68,111,117,118,120,131]. This effect is spectacular in NQO1, in which the CTD of p.P187S is about 17 kJ·mol^−1^ less stable than that of the WT protein [131]. Remarkably, the crystal structure of p.P187S in the NQO1_dic_ state is virtually identical to that of the WT enzyme [69]. This may seem to be paradoxical, because the residue P187 is fully buried in the protein structure, close to the NQO1 dimer interface, and belongs to the minimally stable core of NQO1_apo_ and, thus, a mutation to serine should have catastrophic effects on protein structure and function [132,133]. Detailed biochemical, biophysical, computational and mutational studies have shown that the effects of p.P187S are pleiotropic (i.e., affect many functional features) in the structure, function and stability of NQO1_apo_ and NQO1_holo_, highlighting the critical role of the propagation of the local stability effects due to p.P187S to long distances (over 20 Å) in the protein structure, thus affecting the dynamics and stability of critical regions of NQO1 function [67,68,87,88,111,118,120,129,131,133,134]. These studies on p.P187S have also supported that different functional sites located far (over 20 Å) in the structure are functionally and energetically coupled [131] and, thus, the interaction of NQO1 with its partners could be modulated by an *allosteric* site (i.e., a ligand binding or mutated site) far from the protein:protein binding site (e.g., with p53, p73, HIF-1α…) [63,84,115]). p.P187S is known to lead to the destabilization of NQO1 protein partners. However, it is not clear whether the origin of effects resides in altered PPIs (i.e., binding affinity) or they are simply due to the decreased intracellular stability of this variant that is reflected in those of the partners [67,111,117,118,121,125,131,141]. 

The polymorphism p.R139W represents a beautiful example in which a single nucleotide change may affect protein functionality at post-transcriptional and translational levels. On the one hand, it affects the normal processing of NQO1 mRNA leading to exon 4 skipping, which produces a shorter version of the NQO1 protein that is extremely unstable inside cells due to the lack of part of the catalytic site (residues 102–139) critical for protein folding and stability [139]. On the other hand, full-length NQO1 is also translated, containing the single amino acid exchange p.R139W, which has only mild effects on protein stability and function [78,111,128]. Consequently, the former effect is likely the prevalent one to explain the loss of intracellular NQO1 activity due to this polymorphism [78,111,128,139]. In addition, it must be expected that the decrease in full-length NQO1 protein caused by this polymorphism may negatively impact the stability of protein partners, although to our knowledge, no experimental evidence for this is available. 

The rare mutation p.K240Q, found in the COSMIC database, was later investigated. Additionally, the aim of this study was to determine whether different (i.e., mutational) structural perturbations from glutamine (one of the mildest exchange from lysine) to more perturbing (such as glycine and glutamate) in the CTD could propagate differently through the NQO1 structure to affect multiple functional sites [120,133]. K240 is a solvent-exposed residue involved in a highly stabilizing electrostatic network in the CTD [120]. This site held different types of mutations without compromising the soluble protein levels, activity or thermal stability upon expression in *E. coli* [120,133]. However, detailed biophysical and biochemical experiments actually revealed that the p.K240Q mutation affected the stability of the CTD locally, and this effect was much more pronounced in the more disruptive mutations p.K240G and p.K240E [120,133]. A similar gradual mutational effect was observed on the affinity for FAD, a remarkable result considering that the FAD binding site is located at 20 Å from the mutated site [120,133]. Studies with these mutations thus highlighted the plasticity of the NQO1 protein to respond to mutations in a single site in different ligation states in different functional features (e.g., FAD binding affinity).

Due to this *plasticity* in terms of genotype–phenotype and structure–energetic relationships, we hypothesized that rare mutations found in cancer cell lines (COSMIC) or whole-genome sequence studies (gnomAD) such as p.K240Q would have certain effects on NQO1 protein function and, consequently, on its interaction with protein partners. This would imply that NQO1 function and its ability to develop PPI might be affected in the global human population and, thus, not only related to confirmed cancer cell lines. This concept was recently confirmed with a remarkable outcome [140]. We carried out extensive experimental and theoretical analyses on eight additional mutations, naturally found either in COSMIC or gnomAD databases [140]. The obtained results indicated that mutations found in both databases may display mild to catastrophic consequences in NQO1 function and stability. A plausible consequence of these results is that the PPI developed by NQO1 (for instance with HIF-1α) could be strongly affected in the overall population.

## 3. Targeting the Chaperone Role of NQO1 to Inactivate HIF-1α: Future Perspectives

The chaperone role of NQO1 on HIF-1α has been recently addressed in some detail in relation to hypoxia and normoxia [63,142]. While hypoxic and normoxic conditions provided somewhat different effects, the overall consequence of NQO1:HIF-1α interaction was the stabilization of HIF-1α. The expression levels of NQO1 consistently modulated the levels of *HIF-1α*, likely due to the specific interaction of NQO1 with the HIF-1α ODD domain in the cytosol, rather than to changes in *HIF-1α* mRNA at the transcriptional level [64]. This interaction enhances the intracellular stability of HIF-1α by suppressing its ubiquitination in the cytosol, and thus promoting its nuclear import and transcriptional activity [64]. This chaperone effect is enhanced under hypoxic conditions due to increased expression of NQO1 [64,105]. Intriguingly, the presence of the p.P187S polymorphism did not prevent the chaperone action of NQO1, although this polymorphism is known to reduce NQO1 protein levels due to intracellular destabilization (see Section 2.4).

We could envision different strategies to disrupt NQO1 interaction with HIF-1α leading to increased degradation of the latter: (i)To prevent PPI by targeting the protein:protein binding site or an allosteric site. We must note that allosteric communication of conformational information is well documented and likely a critical feature for the multifunctionality of both NQO1 (Section 2.3 and Section 2.4) and HIF-1α [143,144]. Allosterism in HIF-1α is dramatically exemplified by the negative effector CITED2, which competes with HIF-1α for the same binding site on CBP/p300, attenuating the transcriptional activity of HIF-1α [143,144]. Remarkably, CITED2 and HIF-1α show the same binding affinity for CBP/p300, although the former is much more efficient in displacing the latter from binary complexes with CBP/p300 due to enhanced HIF-1α release linked to the intrinsic disorder of the C-TAD domain. It is worth noting that an important caveat to rational screening for ligands is the lack of high-resolution structural models neither for a complex of NQO1 with a partner nor for the ODD of HIF-1α. There is also no detailed biochemical mapping of the interaction site or plausible molecular models. In addition to identifying non-covalent binders, potential, specific covalent modifiers should also be considered (e.g., the cysteine modification of *Giardia lamblia* triose phosphate isomerase by omeprazole which destabilizes the dimer [145]);(ii)A high-throughput screening for ligands that targets the formation of the NQO1:HIF-1α complex. Both (i) and (ii) would require a rapid, reproducible, robust assay for the interaction in vitro, for example labeling of the proteins with a fluorophore and a quencher. A cell-based assay would also be required to test hits from the screen under in vivo conditions, for example proteins labeled with FRET donors and acceptors, co-immunoprecipitation or indirectly by measuring amounts of HIF1α. Although challenging due to several highly disordered regions in the HIF-1α protein, such screening would still be feasible as recently reported for inhibiting PPI in the cancer-associated and intrinsically disordered protein NUPR1 [146,147];(iii)To use dicoumarol-like molecules that may target that interaction with lower second-site effects. It must be noted regarding this approach that the effects of NADH or dicoumarol have not been tested for the interaction of NQO1 with HIF-1α. Many dicoumarol analogues that function as NQO1 inhibitors have been reported [148,149,150]. It is reasonable to assume that they would also antagonize the NQO1 interactions with binding partners in a similar way to dicoumarol. As such, they represent immediately available compounds that could be tested for their ability to antagonize the NQO1:HIF1α interaction. However, they are likely to inhibit or antagonize many of NQO1′s functions and may, therefore, cause significant off-target effects. Some of these may be undesirable in the context of cancer therapy, e.g., the antagonism of the NQO1/p53 interaction and consequent down-regulation of p53-mediated apoptosis [77,151].

## Figures and Tables

**Figure 1 jpm-12-00747-f001:**
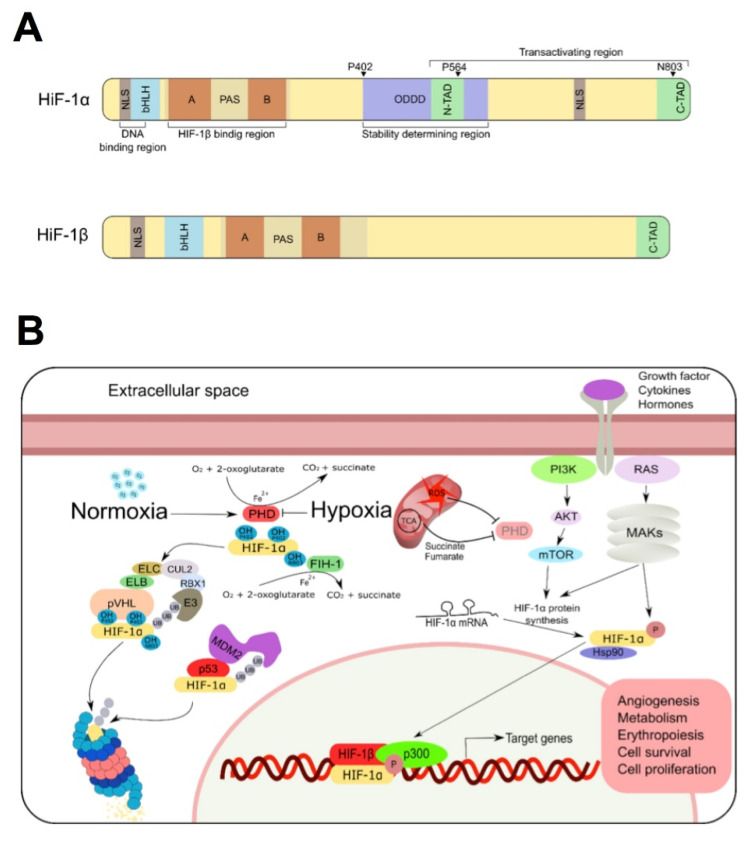
Structure and regulation of HIF. (**A**) Schematic representation of human HIF-1α and HIF-1β structure. Both proteins form part of the HLH–PAS transcription factor family and contain a N-terminal bHLH domain (implicated in DNA binding) and two PAS domains (responsible of its dimerization). HIF-1α contains an oxygen-dependent degradation domain (ODDD) that mediates oxygen-regulated stability, and two transactivation domains (TAD), that mediate its transcriptional activity and its stability. (**B**) In normoxia, HIF-1α is subjected to oxygen-dependent hydroxylation by Prolyl hydroxylase domain (PHD) hydroxylases, conducting HIF-1α to ubiquitination by the von Hippel-Lindau protein (pVHL) and proteasomal degradation. MDM2/p53 are also involved in ubiquitination and degradation of HIF1α protein in a pVHL-independent manner. Under low oxygen levels (as well as some metabolites of TCA cycle), PHD is inhibited, HIF-1α translocates to the nucleus and promotes the activation of target genes. Transactivation of HIF-1α could be also induced by PI3K/MAPKs signaling pathways upon activation of some growth factor, hormones and cytokines.

**Figure 2 jpm-12-00747-f002:**
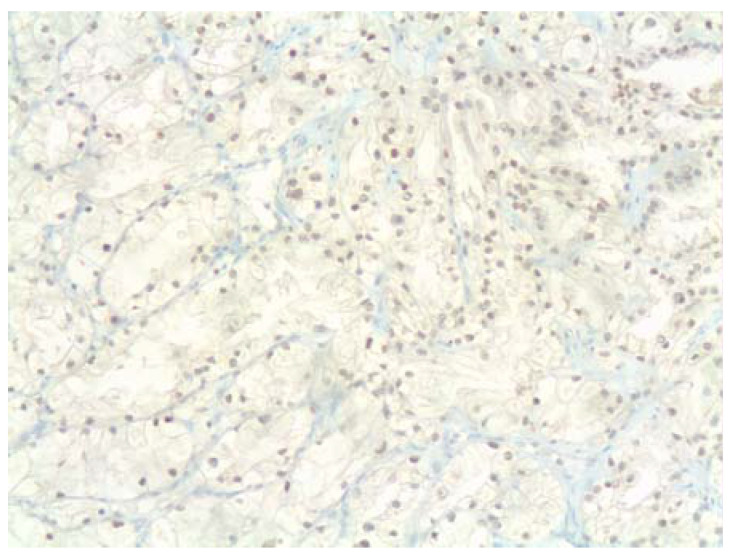
Nuclear HIF-1α (**top**) and NQO1 (**bottom**) expression (brown staining) in serial sections of clear cell renal carcinoma (ccRCC; this particular case carried p.L89L mutation in one allele and a deletion of the other one). Common deletions of the VHL gene result in lack of HIF1 degradation, with the subsequent activation of HIF1 as a nuclear transcription factor that drives tumor cell proliferation. NQO1 is widely expressed in the cytoplasm of tumor cells, including clear cell renal carcinoma, and it might play a role in the postranslational regulation of HIF-1α. Clear cell ovarian carcinoma is another example with strong hypoxic signature where NQO1 might play a role, however we have had no access to cases of clear cell ovarian carcinoma driven by *ARID1A* mutations. Much more work is needed to address this interesting connection. Primary antibodies used: rabbit polyclonal anti-HIF1α (ab2185, dilution 1:100, Abcam, Cambridge, UK) and mouse monoclonal anti-NQO1 (clone A180, dilution 1:600, Thermofisher scientific, Madrid, Spain); peroxidase signal development with Optiview, Ventana.

**Figure 5 jpm-12-00747-f005:**
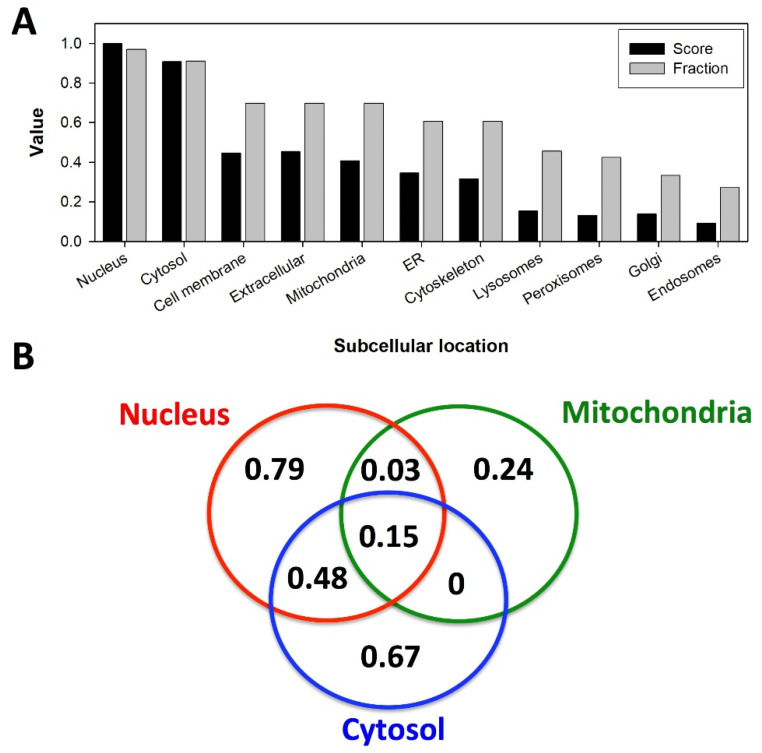
Subcellular location of NQO1 interacting human partners based on data from GeneCards^®^ (https://www.genecards.org/, accessed on 1 April 2022). For all the interactors, the subcellular compartment (location) and its confidence (1–5, from the lowest to the highest) were retrieved from GeneCards^®^. (**A**) Accumulated score for each organelle as the sum of the numerical degree of confidence for all partners found in a given compartment. The highest accumulated score (i.e., for the nucleus) was used to normalize yielding the *Score*. As *Fraction*, we refer to the fraction of all the partners found in a given organelle. Note that the ratio Score:Fraction gives a measure of the degree of confidence for finding a given partner in a given subcellular location. (**B**) Subcellular location of NQO1 partners in the three subcellular locations of NQO1 reported with confidence (i.e., equal or higher than 3) as the fraction of the total of NQO1 partners. Overlapping regions in the Venn diagram reveal the presence of an NQO1 partner in at least two subcellular locations. Numbers in the different regions of the Venn diagram represent the fraction of NQO1 partners present in different subcellular locations.

**Figure 6 jpm-12-00747-f006:**
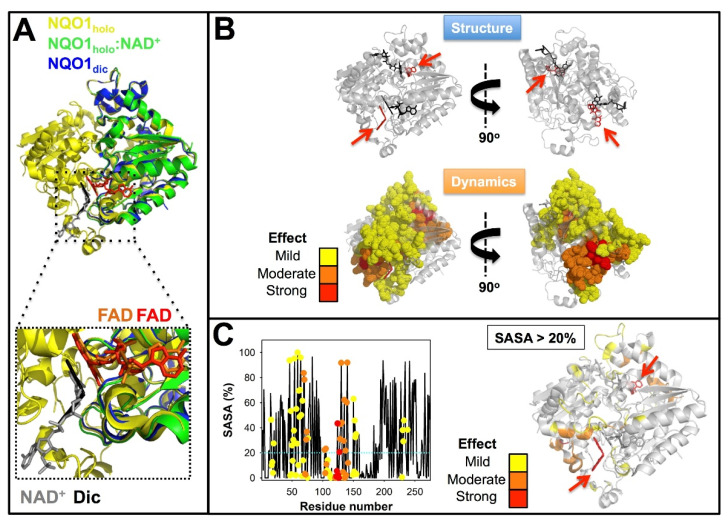
Structure and dynamics of NQO1 upon binding different ligands. (**A**) Structural overlay of the X-ray structures of NQO1_holo_ (1D4A) [81], NQO1_holo_:NAD^+^ (kindly supplied by Profs. Mario Bianchet and Mario Amzel, John Hopkins University Medical School, Baltimore, Maryland, USA) and NQO1_dic_ (2F1O). The lower panel shows a zoom highlighting the position of the FAD (orange, NQO1_dic_ and red, NQO1_holo_:NAD^+^), NAD^+^ (in grey) and dicoumarol (Dic, in black). (**B**) Dicoumarol binding causes long-range effects on the structural dynamics of NQO1 WT. Residues shown in dot representation are those for which the structural dynamics is reduced according to HDXMS [132]. (**C**) Most of the residues whose dynamics are reduced upon dicoumarol binding are solvent-exposed. The plot in the left shows the solvent accessible surface area (SASA) for the each residue as calculated in [132] and color circles indicate the magnitude of the change in structural dynamics. The figure on the right shows the structural location of solvent-exposed residues (SASA > 20%). The color scales in panels B and C reflect the magnitude of the changes in protein dynamics according to [132] and red arrows indicate the position of dicoumarol.

## Data Availability

All primary data will be provided upon appropriate request.

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
