# Peer review of "Targeting HIF-1α Function in Cancer through the Chaperone Action of NQO1: Implications of Genetic Diversity of NQO1"

_jpm, 2022, doi:10.3390/jpm12050747_

Round 1

Reviewer 1 Report

I would like to thank the authors for addressing my comments.

Reviewer 2 Report

I have read the comments and the manuscript carefully again. I have no further comments.

This manuscript is a resubmission of an earlier submission. The following is a list of the peer review reports and author responses from that submission.

Round 1

Reviewer 1 Report

This well-written manuscript of Salido et al. (2021), titled “Targeting HIF-1α Function in Cancer through the Chaperone Action of NQO1: Implications of Genetic Diversity of NQO1”, featured two cornerstones. The first is HIF, mainly HIF1α, with all pertinent information involving structure, functions, and regulations, emphasizing stability. The second is NQO1, wherein the authors provided a plethora of information on various aspects of NQO1, such as regulation, activity, molecular determinants of NQO1 as a chaperone of many different proteins, etc. The authors highlighted the potential therapeutic utility of targeting the intracellular stability of HIF-1α  by acting on its chaperone, NQO1. This review provided crucial knowledge and a much-needed platform for a better understanding of NQO1 as a therapeutic target, which, indeed, necessitates rigorous investigation. However, some issues need to be addressed, as detailed below.

  • The authors mainly focused on HIF1α stabilization, whereas it has been frequently reported that HIF1α translation is estimated to account for 40%–50% of HIF1α induction under hypoxia/stress (e.g PMID: 17967866, PMID: 16507764, PMID: 25965573), and that cap-independent translation seems to play an essential role in HIF1α expression. The authors are encouraged to address this crucial regulatory mechanism, especially since there are emerging new strategies targeting specific elements of the translational machinery, such as eIF4A, eIF4E:eIF4G interaction, for innovative cancer therapy (e.g PMID: 29293271). Targeting HIF1α stability through NQO1 and translation might represent a very promising strategy.

  • The authors discussed ccRCC as a model of LoF mutations at pVHL with subsequent HIF1α stabilization. However, extensive evidence supports a critical role for HIF2α in ccRCC and represents a potential therapeutic target in ccRCC (e.g PMID: 32807776, PMID: 33868900, PMID: 29938199). The authors are encouraged to briefly address this overlooked part of HIF2α. Further, the authors showed HIF1α expression in ccRCC; it will be solid supportive data to show NQO1 expression in ccRCC.

  • Clear cell carcinoma of the ovary shares striking similarities with ccRCC, and is characterized by a strong hypoxic signature and frequent loss of ARID1A similar to uterine cancers. Previous work has shown that NQO1 expression is significantly higher in ARID1A-mutant tumors than ARID1A-wildtype tumor using TCGA uterine cancer data set (source: https://digitalcommons.library.tmc.edu/cgi/viewcontent.cgi?article=1682&context=utgsbs_dissertations). The authors are encouraged to highlight this exciting link involving ARID1A-NQO1 for potential therapeutic targeting.

  • The authors are encouraged to address the controversial roles of NQO1 in cancer. Although NQO1 expression is associated with an aggressive course and poor prognosis in many cancers, it has been shown to drive AMPK activation and induce cancer cell death under oxygen-glucose deprivation (PMID: 25586669).

Reviewer 2 Report

The review of Salido et al. discuss the potential of targeting HIF1a function in cancer through the chaperone action of NQO1. The chosen topic is interesting, but the review needs a major revision. Due to the length of the review, the focus is unfortunately somewhat lost. In particular, the first chapter, which focuses on HIF1a, should be shortened and concentrate on the things that really lead to the topic. Similar problems exist with the other parts. It should be critically checked again, which information is really necessary for the consideration of the topic. Furthermore, attention should be paid to the scientific accuracy. The following comments are just a few examples.

Furthermore, the role of HIF2a is completely ignored. Especially in tumors with VHL mutations, as mentioned in the review ccRCC, but also for example pheochromocytomas and paragangliomas, HIF2a is the factor associated with increased tumor progression and aggressiveness [e.g. 1-3]. This is why many current discussions focus more on addressing HIF2a stabilization. This should at least be discussed. 

The manuscript also needs a thorough linguistic revision.

[1] Bechmann, N., & Eisenhofer, G. (2021). Hypoxia-inducible Factor 2α: a key player in tumorigenesis and metastasis of pheochromocytoma and paraganglioma?. Experimental and Clinical Endocrinology & Diabetes.

[2] Courtney, K. D., Ma, Y., de Leon, A. D., Christie, A., Xie, Z., Woolford, L., ... & Brugarolas, J. (2020). HIF-2 complex dissociation, target inhibition, and acquired resistance with PT2385, a first-in-class HIF-2 inhibitor, in patients with clear cell renal cell carcinoma. Clinical Cancer Research, 26(4), 793-803.

[3] Chen, W., Hill, H., Christie, A., Kim, M. S., Holloman, E., Pavia-Jimenez, A., ... & Brugarolas, J. (2016). Targeting renal cell carcinoma with a HIF-2 antagonist. Nature539(7627), 112-117.

Specific comments:

Throughout the manuscript, it is not clear when you refer to the protein and when the gene. Please use the common notations, e.g., cursive vs. non-cursive. When you talk about pVHL, do you mean the protein? But then you must not use this, for example, when you talk about VHL mutations (L182).

L38-39: “with heterodimers known as hypoxia factor” --> HIF1-3α dimerize with ARNT/HIFβ --> be more specific and also add α beyond HIF-1-3

L40-42: HIF2a regulate similar processes and needs to be included

L53: Is it really cancer development? Progression? Please specify

L115: “HIF1a is induced by numerous stimuli other than hypoxia” --> I think you are talking about the gene expression, right (not cursive)? If so, the sentence is not correct, because hypoxia regulate the stabilization of HIFs and cytokines and hormones affect the gene expression. Please correct.

L1676-169: It is true that pseudohypoxia plays a role in serveral tumors (see also [1]), but please define it correctly and describe the mutations that can lead to this phenotype (e.g. specific mutations).

L170-185: As mentioned above, gen vs. protein. In addition, the literature should be critically reviewed in this context to determine whether HIF1 or HIF2 is really important.

L207-210: Sentence needs to be carefully restructured

Figure 2: Own staining? Antibody used? Is a VHL patient presented? Is it possible to compare a staining of the VHL patient with a patient without VHL mutation?

L222: “other types of cancer” What are you referring to here?

L235: “could”--> delete

L242-245: “Since our knowledge on this particular PPI is very limited…” If you are not an expert in this area, you may want to have someone else write this review, or you may want to seek appropriate assistance.

L352: Is there something known for HIF2a? HIF1a and HIF2a have overlapping and distinct target genes. If its not HIF1/2 related how do you explain the upregulation under hypoxic conditions?

Minor comments:

Please replace symbols for example L18, L239, L617 and others

L183: Fumarate Hydroxylase --> fumarate hydroxylase

L220, L351: HIF1A --> HIF1α

L328: mediator (small)